


# An integrated hydrological and hydraulic modelling approach for the flood risk assessment over Po river basin.

Rita Nogherotto[1], Adriano Fantini[1], Francesca Raffaele[1], Fabio Di Sante[1], Francesco Dottori[2], Erika Coppola[1], and Filippo Giorgi[1]

[1]International Centre for Theoretical Physics,Trieste, Italy
[2]European Commission, Joint Research Centre, Ispra, Italy

## Abstract

Identification of flood prone areas is instrumental for a large number of applications, ranging from engineering to climate change studies, and provides essential information for planning effective emergency responses. In this work we describe an integrated hydrological and hydraulic modeling approach for the assessment of flood-prone areas in Italy and we present the first results obtained over the Po river (Northern Italy) at a resolution of 90m. River discharges are obtained through the hydrological model CHyM driven by GRIPHO, a newly-developed high resolution hourly precipitation dataset. Runoff data is then used to obtain Synthetic Design Hydrographs (SDHs) for different return periods along the river network. Flood hydrographs are subsequently processed by a parallelized version of the CA2D hydraulic model to calculate the flow over an *ad hoc* re-shaped HydroSHEDS digital elevation model which includes information about the channel geometry. Modeled hydrographs and SDHs are compared with those obtained from observed data for a choice of gauging stations, showing an overall good performance of the CHyM model. The flood hazard maps for return periods of 50, 100, 500 are validated by comparison with the official flood hazard maps produced by the River Po Authority (Adbpo) and with the Joint Research Centre's (JRC) pan-European maps. The results show a good agreement with the available official national flood maps for high return periods. For lower return periods the results and less satisfactory but overall the application suggests strong potential of the proposed approach for future applications.

Keywords: Flood hazard; Flood mapping; CHyM hydrologic model; CA2D hydraulic model.



# 1 Introduction

The last few decades have seen increased interest towards the study of floods, their conse-
quences and the development of measures to reduce their impact. Flood hazard maps are
designed to indicate the probability and/or magnitude of inundations over a given area and
are used as an important decision making tool for multiple purposes ranging from infras-
tructure development to disaster response planning. This is also endorsed by the European
Union Flood Risk Management Directive (European Commission, 2007), which mandate is
the development of flood hazard maps for exposed territories, showing the potential con-
sequences associated with different flood scenarios, in order to guarantee an effective basis
for technical, financial and political decisions regarding the flood risk management. Until
recently, flood hazard maps were only available for few regions of the globe, and with coarse
resolutions, due to the high data and computational requirements of the hydraulic models
employed in their production (Moel et al., 2009). The increase of computational power and
the availability of remotely sensed datasets, however, have made the application of flood
models with higher resolution (less than 1 km) possible even over large domains (Wood
et al., 2011).

Different methods to quantify flood hazard can be employed, resulting in different types
of flood maps (Moel et al., 2009). Within the different approaches, the common steps are
essentially two: 1) the estimation of the discharges for specific return periods and 2) the
combination of the discharges with a digital elevation model (DEM) for the creation of the
flood map.

For limited area gauged basins, where discharges data are available, the first step can be
accomplished by using frequency analyses on discharge records and fitting extreme values
distributions (e.g. Te Linde et al., 2008). For larger domains, flood information can be ex-
trapolated to ungauged areas using regionalisation techniques (e.g. Merz and Blöschl, 2005)
or by using hydrological models to calculate discharges (Bárdossy, 2007; Khan et al., 2011).
These models require spatially explicit meteorological (e.g. temperature, precipitation, evap-
oration, radiation), soil, and land cover data as input and they solve the water balance for
each geographical unit for each time step, to yield the discharges for all river stretches. The
strength of this approach is not only the applicability over ungauged regions, but also the
possibility of assessing the impact of changes in climate and/or land cover on floods. The sec-
ond step is usually accomplished by using hydraulic models specifically designed for solving
channel and floodplain hydraulic routing. Historically, this was usually performed by mod-
eling fluvial hydraulics with one-dimensional finite difference solutions of the full St. Venant
equations (see Fread, 1985; Samuels, 1990), using models such as MIKE11 (Havnø et al.,
1995) and HEC-RAS (Brunner, 2002). These schemes describe the river channel and flood-
plain as a series of cross sections perpendicular to the flow and estimate average velocity and
water depth at each cross section. Despite the successful validation of flood inundation ex-
tent using low resolution satellite imagery (Bates et al., 1997), the one-dimensional schemes
have the drawbacks of being computationally expensive and the areas between the cross
sections are not explicitly represented (Samuels, 1990; Bates and De Roo, 2000). Thanks to





the increasing availability of high resolution Digital Elevation Models (DEM) for floodplain areas, two-dimensional distributed models have been developed to allow a better conjunction with the elevation of the channel and of the floodplain surface, and to guarantee the calculation of the water depth and depth-averaged velocity at each computational node at each time step. Examples of such two-dimensional schemes are LISFLOOD-FP (Bates and De Roo, 2000), RBFVM-2D (Zhao et al., 1994) and TELEMAC-2D (Galland et al., 1991). These physically based models solve the Shallow Water Equations (SWEs) and, due to the recent advancement in parallel computing techniques, can be applied over large areas at high resolution. In recent years, a new approach was developed which employs cellular automata (CA) algorithms instead of directly solving the SWEs for each interface: for each timestep, the new state of a cell depends only on the state of the neighbouring cells at the previous timestep, according to a set of rules. This technique allows to model complex physical systems using simple operational rules (Wolfram, 1984), drastically reducing the computational requirements compared to physically based models. These algorithms are therefore well suited for parallel computation and have been successfully used to simulate many types of water related problems (e.g. Coulthard et al., 2007; Krupka et al., 2007; Austin et al., 2013).

An example is the CA2D model developed by Dottori and Todini (2011). The CA2D model uses a 2D cellular automata approach and the equations developed for the LISFLOOD-FP model (Bates et al. (2010)) to make high resolution simulations possible at continental and global scale (Dottori et al. (2016d)).

In this study we describe an integrated hydrological and hydraulic modelling approach which uses the Cetemps Hydrological Model (CHyM, Coppola et al. (2007)) and a modified version of the CA2D hydraulic model, hereinafter referred to as CA2D$_{par}$. CA2D$_{par}$ includes a parallel algorithm with the physics of the CA2D model but that can be run with multiple processors to further speed up the computation. Furthermore, to better represent river flow and flooding processes, we produced a re-shaped digital elevation model which includes information about the channel geometry by simulating a "digging" assuming that discharges associated to return periods of 1.5 years produce no floods as they represent the conveyance capacity of the river channel. This model has been used over the entire Italian territory. In the present work we focus on the results obtained over the Po river, which is the river with the largest average daily discharge in the Italian peninsula and in whose basin 40% of the gross domestic product of Italy is produced (Montanari, 2012).

In Section 2 we will describe the observational and modelled data and the method applied for flood hazard assessment of the western basin of the river Po. Section 3 will present the results, by means of a validation of the obtained SDHs, a validation of the hazard maps against observations and against existing flood hazard maps.




## 2 Data and methods

The approach proposed herein assumes that large scale flood hazard maps can be derived from an ensemble of small scale simulations of flood processes, arranged to cover the entire river network, as previously demonstrated in literature (Alfieri et al., 2013, 2014; Dottori et al., 2016d). The procedure is composed by the following steps: 1) the hydrological simulations are setup and calibrated for the production of a long-term discharge time series; 2) the designed hydrographs are derived for different selected return periods; 3) the floodplain hydraulic simulations are performed and the flood maps for each return period are produced. These three different steps will be described in detail in the following subsections.

### 2.1 The observational data and the hydrological model CHyM

Hydrological simulations are performed using the CETEMPS Hydrological Model (CHyM) (Coppola et al., 2007), the distributed hydrological model developed by the CETEMPS Center of Excellence at the University of L'Aquila. CHyM uses information from a Digital Elevation Model (DEM) and produces a D8 connected river network, using cellular automata algorithms to resolve local singularities and no-flow points (Coppola et al., 2007).

Input precipitation from various sources can be assimilated, including gridded precipitation from observations and models. Discharge is routed through each grid cell using continuity and momentum equations based on the kinematic shallow water approximation of Lighthill and Whitham (1955). CHyM is specifically designed for Italian river catchments and has been widely tested for a variety of regions across Italy, and in particular for the Po basin (Coppola et al., 2014; Verdecchia et al., 2009; Tomassetti et al., 2005b). For this study, nine separate domains are simulated, with a resolution varying between 300 and 900m (Fig. 1). The domains are matching the operational domains simulated by CETEMPS to forecast potential floods using stress indexes (Tomassetti et al., 2005a; Verdecchia et al., 2008), but they are higher resolution because the HydroSHEDS Digital Elevation Model is used (Lehner et al., 2013), which is specifically conditioned for hydrological usage. The choice of the DEM is crucial to ensure correct river routing especially in large, flat areas such as the Po plain. The simulations span the period 2001–2016 and are driven by the newly-developed hourly precipitation dataset GRIPHO (Fantini et al., 2019; Fantini, 2019), which includes quality-controlled data from 3712 precipitation stations covering all of Italy. MM5 weather forecasts (Grell et al., 1994), operationally in use at CETEMPS for more than 20 years (see e.g. Bianco et al., 2006), are employed to fill data gaps in GRIPHO.

Further information on the hydrological simulations used for this study, including validation against discharge observations, can be found in Fantini (2019, chapters 4 and 5).




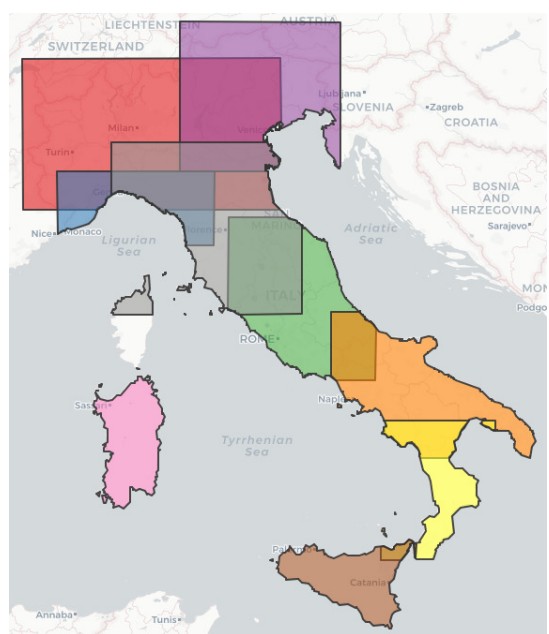

Figure 1: The nine domains on which the CHyM model is run operationally.

## 2.2 Processing the hydrological inputs: the Synthetic Designed Hydrographs (SDHs)

The statistical procedure applied in this study is based on the work of Maione et al. (2003), who performed a Flood Frequency Analysis (FFA) starting from observational data for the Po river basin. The aim is to obtain curves describing the typical discharge timeseries of the event at that river point for the given Return Period. These $Q_{RP}(t)$ curves will be called Synthetic Design Hydrographs (SDHs) and they represent the discharge ($Q$) of a typical extreme event as a function of the Return Period ($RP$) and the time ($t$). SDHs are estimated and used as input data for the hydraulic model in order to predict the corresponding maximum flood inundation extent and depth (see subsection 2.3). Simulations were performed using observational data described in subsection 2.1 and processed to derive synthetic flood hydrographs throughout a statistical analysis of the Flow Duration Frequency (FDF) reduction curves $Q_D(RP)$(Maione et al., 2003) . These curves represent the typical discharge with Return Period $RP$ averaged over any duration $D$ around the flood peak. For each station along the river network $Q_D(RP)$ can be calculated from statistical analyses of historical hydrographs. Similarly to the work of Maione et al. (2003) we used the empirical relationship proposed by NERC (1975) defining the reduction ratio ($\epsilon_D$), which is the ratio of the FDF

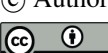



and the peak flood discharge ($Q_0(RP)$), as follows:

$$\epsilon_D(RP) = \frac{Q_D(RP)}{Q_0(RP)}. \tag{1}$$

In this work we assume $\epsilon_D$ is independent on the return period, which occurs for medium-large catchments, as done by Maione et al. (2003) and Alfieri et al. (2013). When performing the calculation of the FDF around each historical flood peak, the centre of the duration window of width D is chosen as to maximise the average computed discharge $Q_D$:

$$FDF = Q_D = \frac{1}{D}\max\int_t^{t+D} Q(\tau)d\tau, \tag{2}$$

where $t$ and $\tau$ represent time. The shape of the final synthetic hydrograph will be determined by the peak-duration ratio $r_D$ that is the ratio of the time before the peak and the total duration $D$ of the averaging window. The smaller the $r_D$, the more skewed the hydrograph will be towards steeper (flatter) rising (falling) limbs of the hydrograph. Centring on $t = 0$ the peak flood timing, the two limbs of the hydrograph can be described as:

$$\int_{-r_DD}^{t=0} Q(\tau) = r_D D Q_D(RP) \tag{3}$$

and

$$\int_{t=0}^{(1-r_D)D} Q(\tau) = (1-r_D)D Q_D(RP), \tag{4}$$

where $Q_D(RP)$ is the typical FDF curve for the Return Period $RP$. The construction of the SDH is performed imposing that the maximum discharges for each duration coincides with the value obtained from the FDF curves, in a given duration $D$ for each value of the return period $RP$. Thus the SDH is obtained differentiating with respect to the duration $D$, obtaining for the falling limb:

$$SDH = Q_t(RP) = \frac{d/dD[(1-r_D)DQ_D(RP)]|_{D=D(t)}}{d/dD[(1-r_D)D]|_{D=D(t)}} \tag{5}$$

where $t = (1-r_D)D$.

The maximum flood discharge $Q_0(RP)$ for any given Return Period $RP$ must then be calculated by fitting an appropriate extreme distribution. Following Alfieri et al. (2015) and Maione et al. (2003), we chose the Gumbel distribution, so that:

$$Q_0(RP) = u - \alpha\ln\left[-\ln\left(1 - \frac{1}{RP}\right)\right], \tag{6}$$

where the parameters $u$ and $\alpha$ are estimated from the fit, and are used for the differentiation of the equation 5. The equation, representing the falling limb of the SDH, allows us to calculate a typical flood event discharge timeseries for any location and Return Period, starting only from the timeseries of yearly maximum discharges. Further details about the procedure and its implementation can be found in Fantini (2019). Figure 2 shows SDHs for seven Return Periods obtained applying the procedure described in section 2.2 for a station on the Tanaro river, a tributary of the Po river.


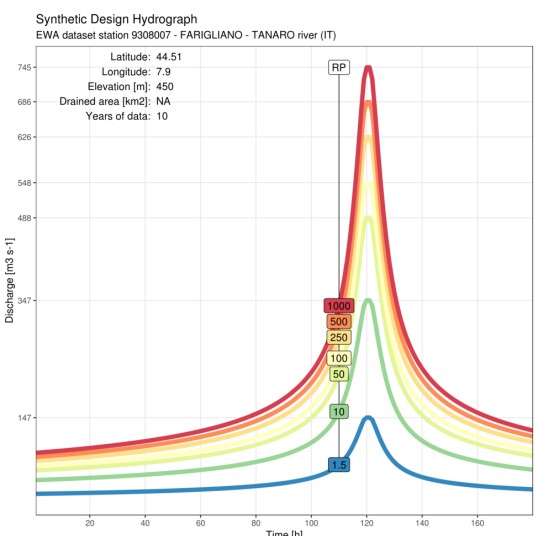

Figure 2: Example Synthetic Design Hydrograph computed following the procedure described in section 2.2 for a station on the Tanaro River, tributary of the Po river. Seven Return Periods (1.5, 10, 50, 100, 250, 500 and 1000 years) are shown.

## 2.3 Modelling the flood inundation: the hydraulic model

Floodplain hydraulic simulations are performed with a modified version of the 2D hydraulic cellular automata model CA2D. The model, described and validated in Dottori and Todini (2011), is based on a simple cell-centred finite volume scheme, which uses the Euler explicit scheme for the integration in time. The momentum equation is solved for each time step, computing volume exchanges between grid cells along the cell's borders. Volumes of each cell are successively updated using volume conservative equations. For this study, the model is run using the semi-inertial formulation of the momentum equation (Bates et al. (2010)), which allows to reproduce channel and floodplain flow processes with a good level of detail with a considerably reduced computational effort (Dottori and Todini, 2011).

The model version CA2D$_{par}$ has been written using Fortran90 standard and it has the original model described in Dottori and Todini (2011) as a starting point. The physics is represented on a cartesian 2D grid that allows a good level of scalability. The parallel code has been carried out using the message passing interface (MPI) communications. A number of subroutines has been introduced in the code to deal with the parallelization and are compiled as separated modules. The parallelization of the code increases as expected the performance of the model which is up to 7.5 times faster respect to the original, even with a limited number of cores (Fig. 3).

The flood inundation extent is dependent on the spatial extent of the performed hydraulic simulations, and it is therefore important to define the number and location of the hydraulic





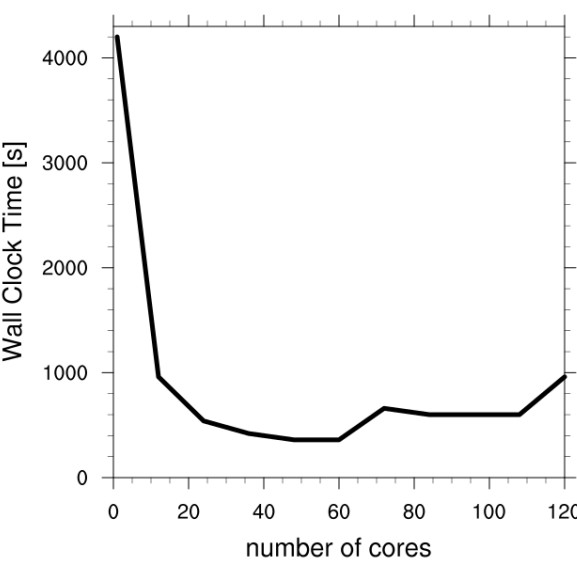

Figure 3: Wall-clock time (s) variation with the number of cores achieved with the parallelization of the CA2D model.




simulation in order to achieve the full coverage of the interested river network. The following section will show the results obtained and it is organised in three steps: 1) calculation of the design flood hydrographs for the available observational stations along the river network using observational data, 2) calculation of the design flood hydrographs obtained using the CHyM model data on the same locations, and comparison of the two series of hydrographs for a validation of the hydraulic model along the Po river, 3) calculation of the design flood hydrographs in selected points along the river network at regular distance from each other and performance of the CA2D$_{par}$ simulations using the SDH as inputs.

## 2.4    The production of the flood maps.

Currently the Shuttle Radar Topography Mission (SRTM) digital elevation model (Farr et al., 2007; Rabus et al., 2003) is considered as one of the best openly available data set for flood modeling offering near-global converage (Hirt et al., 2010; Jing et al., 2014). The void-filled HydroSHEDS variant of SRTM was used in this work with 3 arc sec resolution (Lehner et al., 2006, 2008).
As described in Neal et al. (2012) and Sampson et al. (2015) the inclusion of a river channel network is necessary to guarantee acceptable results in the simulation of flood depths and extent. River widths and depths are however difficult parameters to estimate as it is not possible to measure them remotely on large scales. Natural and artificial river defenses are also challenging to incorporate as their features are smaller than the model grid resolution (Sampson et al., 2015). Moreover their spatial distribution on large scales is not available as literature about fluvial flood defenses generally refers to individual sites (e.g. Brandimarte and Di Baldassarre, 2012; Te Linde et al., 2011). Available remotely sensed data were recently used to generate regional to global estimates of river widths and depths (Andreadis et al., 2013; Gleason and Smith, 2014) by coupling river network data to web based imagery services such as Google maps or Bing maps.
In this study we have used the near-global database of bankfull depths, based on hydraulic geometry equations and the HydroSHEDS hydrography data set described in Andreadis et al. (2013), to estimate the channel conveyance. The idea is to link the channel geometry to the discharge return period, as it guarantees that channels, properly sized, are able to contain the simulated flows and moreover mitigates against the problem of missing information about the river banks. We have used the river bankfull depths information to reshape the HydroSHEDS digital elevation model by assuming a bankfull discharge return period of 1.5 years (Leopold, 1994; Harman et al., 2008; Andreadis et al., 2013; Sampson et al., 2015; Neal et al., 2012). In order to include information about the geometry of the river, the natural and man-made banks, we used the bankfull depths to artificially "dig" the HydroSHEDS DEM until we obtained a no-flood map correspondent to the return period of 1.5 years, which represents the conveyance capacity of the river channel.

As stated in 2.1 a 15-years continuous discharge time series with Italian coverage is generated using the CHyM hydrological model from January 2001 to December 2016. Floodpeaks with 50, 100, 500 year return period are derived for each river point in the model and downscaled




to the river network at 3 arc sec resolution. Design flood hydrographs are then used to
perform small scale floodplain hydraulic simulation on points which will be hereafter referred
to as "virtual stations" (see Fig. 4), located every 10 km along the river network, for rivers
with drainage areas larger than A=5 km$^2$, using the hydraulic model CA2D$_{par}$. For each
virtual station the simulation was run over a sub-domain, 0.3°× 0.3°, chosen to optimise the
computational effort, as the simulation time is strongly affected by the size of the domain.
For each return period a total of 474 simulations were performed and merged to produce a
Western Po river flood hazard map (Fig. 5).

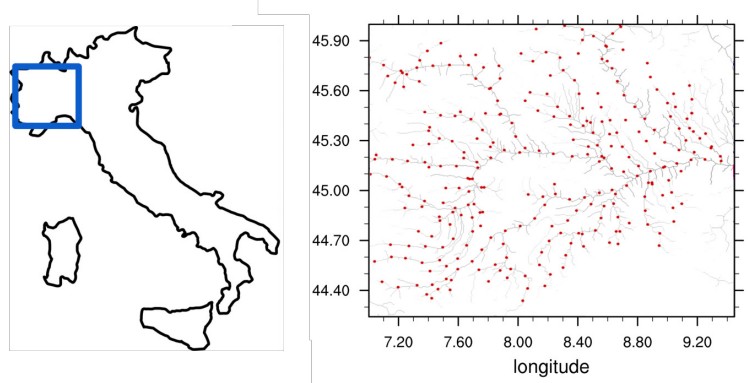

Figure 4: Virtual stations selected for drainage areas larger than A=5 km$^2$ and regularly
spaced every 10 km along the high-resolution river network of the analyzed domain (blue
box on the left).

## 3 Results

### 3.1 Validation of the SDHs.

Tuning and testing of the method were performed on the upper Po basin, due to previous
experience with the hydrological model on this domain (Coppola et al., 2014), availability of
reliable observed discharge data, and lack of large water management structures. Due to the
relatively small size of the simulated domains, the duration of all flood simulations was set
to 240 $h$. The SDHs were validated using data from the CHyM model and observations from
31 gauge stations along the Po river. Figure 6 shows the results of the comparison between
the SDHs obtained with observational data and those obtained with modelled data. The
SDHs are generally closely approximated by the model, both in the peaks and in the area
of the curves. The coefficient of determination ($R^2$) is 0.85 for the SDHs areas and 0.92 for
the SDHs peaks which are the same values reported in Rojas et al. (2011) for a hydrological
model of Europe without bias correction of climate data and in Paprotny et al. (2017).





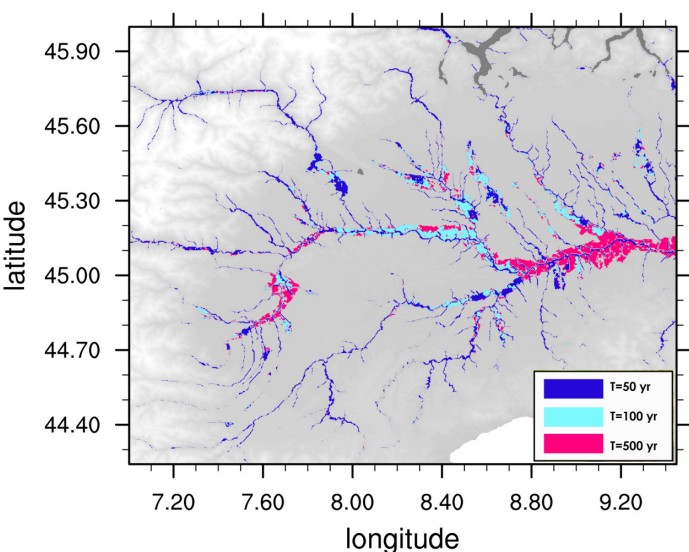

Figure 5: Western Po river flood hazard map for the Return Periods of 500, 100 and 50 years.

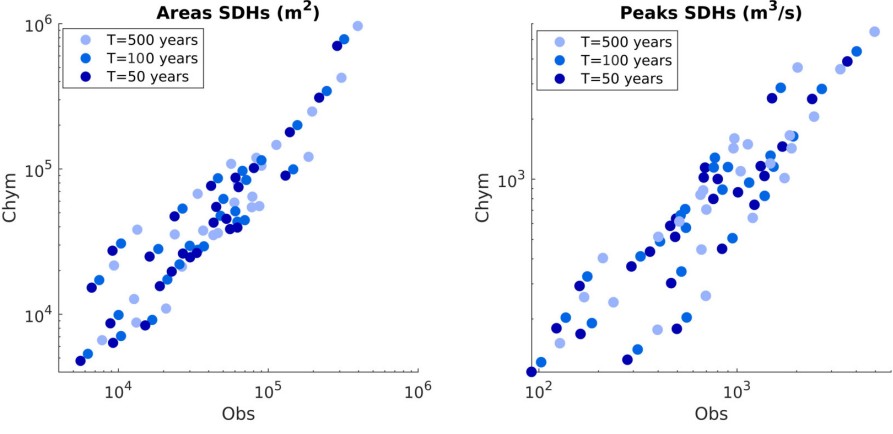

Figure 6: Comparison of simulated (CHyM) and observed (Obs) SDHs areas (a) and discharges peaks (b) for 31 gauge stations along the Po river, for three return periods.





## 3.2 Comparison against observations: a case study

Validation of flood hazard models is achieved trough the evaluation of the model accuracy
in estimating the probability of flood occurrence and the evaluation of relevant hazard vari-
ables of an event (e.g. flood extent and depth, flow velocity). Unfortunately the evaluation
is strongly limited by the scarce availability of reference flood maps and flood observations
and is a key topic in flood risk analysis. Various methods were suggested by previous stud-
ies. One consists in comparing the produced maps with previous maps based on statistical
estimation of peak discharges (Pappenberger et al., 2012); another method performs a qual-
itative assessment of the flood events against satellite flood images (Rudari et al., 2015).

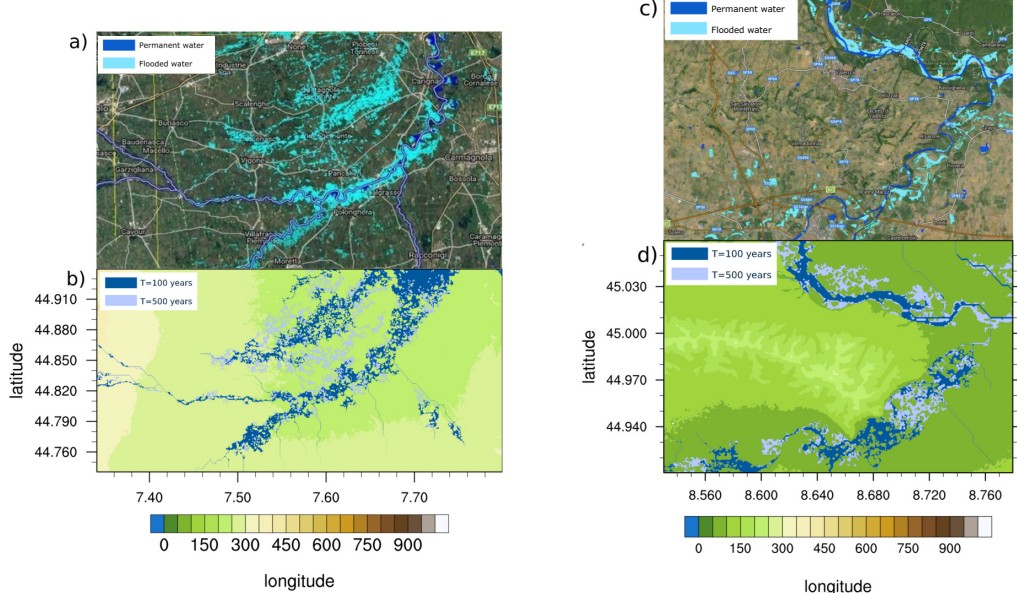

Figure 7: Case studies in November 2016, used for the validation of the method: panels above show floods as acquired by the satellite COSMO-SkyMed (COSMO-SkyMed Image ©ASI (2016). All rights reserved). Panels below show floods as modelled by the integrated CHyM-CA2D$_{par}$ method. Panels (a) and (b) show flooded areas in the south of Turin. Panels (c) and (d) show flooded areas in the area of Alessandria.

In order to perform a first validation of the flood hazard mapping methodology we consider a
case study of a flood recently occurred in Northern Italy, catalogued as an event with return
period of 100 years. November 2016 was characterized by a heavy rainfalls event involving the
territory of North West of Italy, in particular the Regions of Piemonte and Liguria. The bad




<sub>289</sub> weather conditions and the persistence of precipitations caused the increase of hydrometric
<sub>290</sub> levels of all the rivers in particular in the Po river basin.

<sub>291</sub> Figures 7 (a) and (c) show the images from satellite COSMO-SkyMed (CSK) (Covello et al.,
<sub>292</sub> 2010), a four-satellite constellation which gives the possibility of acquiring X -band Syn-
<sub>293</sub> thetic Aperture Radar (SAR) data day and night, regardless of weather conditions and is
<sub>294</sub> fully operational since the 2008. It provides radar data characterized by short revisit time
<sub>295</sub> and therefore useful for flood mapping evaluation. The lower panels show the flood maps
<sub>296</sub> corresponding to two different return periods (T=500 and T=100 years). We can see that
<sub>297</sub> the observed event, associated to a return period of 100 years, is fairly good represented by
<sub>298</sub> the model (Fig. 7 (b) and (d)) as the maps include the particular events observed.

## <sub>299</sub> 3.3  Comparison against existing flood hazard maps

<sub>300</sub> Another approach for the validation is to perform an evaluation against existing high-
<sub>301</sub> resolution flood hazard maps (Alfieri et al., 2013; Sampson et al., 2015; Winsemius et al.,
<sub>302</sub> 2016). The evaluation of simulated flood maps against reference maps is performed using
<sub>303</sub> the indexes proposed in literature (Dottori et al., 2016d; Bates and De Roo, 2000; Alfieri
<sub>304</sub> et al., 2014). The Hit Ratio index (HR), defined as:

$$\mathrm{HR} = (F_m \cap F_o)/(F_o) \qquad (7)$$

<sub>306</sub> evaluates the agreement of modelled maps $(F_m)$ with existing maps $(F_o)$. This index does
<sub>307</sub> not take into account the overprediction and underprediction of the flooded area, therefore
<sub>308</sub> two other measures are calculated to account for this: the False Alarm index (FA), defined
<sub>309</sub> as

$$FA = [F_m - (F_m \cap F_o)]/(F_o) \qquad (8)$$

<sub>311</sub> where $F_m - (F_m \cap F_o)$ is the flooded area wrongly predicted by the model, and the Critical
<sub>312</sub> Success index (CS), defined as:

$$CS = (F_m \cap F_o)/(F_m \cup F_o). \qquad (9)$$

<sub>314</sub> The produced flood hazard maps, hereinafter referred to as "CA2D maps", are tested against
<sub>315</sub> the official hazard AdbPo flood maps (http://www.adbpo.gov.it), produced by the River Po
<sub>316</sub> Authority, who classifies the flood plain of the Po river into three levels corresponding to
<sub>317</sub> return periods of 20-50 years (high frequency), 100-200 years (medium frequency) and 500
<sub>318</sub> years (low frequency).
<sub>319</sub> In addition, we compare the CA2D maps with the flood hazard maps produced by the Joint
<sub>320</sub> Research Centre of the European Commission (JRC). The JRC maps are freely available on-
<sub>321</sub> line and are based on streamflow data from the European Flood Awareness System (EFAS
<sub>322</sub> (Demeritt et al., 2013) and also calculated with a spatial resolution of 3" (Dottori et al.,
<sub>323</sub> 2016a,b,c). To perform the indexes calculations, we have focused our analysis on a smaller


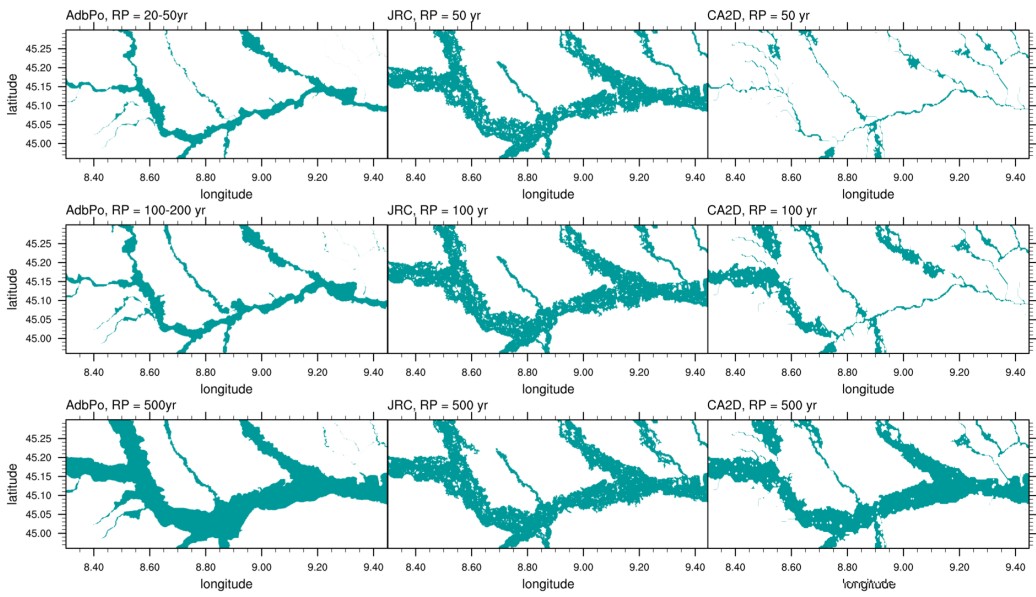

Figure 8: Adbpo, JRC and CA2D flood hazard maps for the 50 years return period (upper panels), 100 years return period (central panels) and 500 years return period (lower panels)

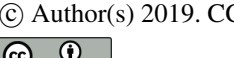


³²⁴ portion of the domain, centred on the main river, removing flooded areas originating from
³²⁵ river sections with an upstream area smaller than 500 km² since they are not simulated
³²⁶ and therefore not included in the JRC maps. The JRC flood maps used for the compari-
³²⁷ son do not consider flood defences and river geometry, for this reason we only calculate the
³²⁸ performance indices (Eq. (7), (8) and (9)) for the 500 years return period, reported in Ta-
³²⁹ ble 1. Indices are calculated for the CA2D and JRC maps ($F_m$) against the Adbpo maps ($F_o$).
³³⁰

|      | Hit Rate | False Alarm | Critical Success |
|------|----------|-------------|------------------|
| JRC  | 0.83     | 0.15        | 0.73             |
| CA2D | 0.76     | 0.12        | 0.67             |

Table 1: Evaluation of the CA2D and JRC flooded extent against official flood hazard maps
(Adbpo) for thre return period of 500 years.

³³¹ As can be seen, the CA2D maps provide fairly good results for the 500 years return period,
³³² with a HR of 0.76, a CS index of 0.67 and a very low false alarm value (0.12), while results
³³³ are less satisfactory for lower return periods, with considerable underestimation of flood
³³⁴ extent respect to the offical maps (see Fig. 8). JRC maps also show fair results for the 500
³³⁵ years return period, with a HR of 0.83, a CS of 0.73 and FA of 0.15, and are similar to
³³⁶ CA2D maps (Fig. 9), but they systematically overestimate flood extent for the lower return
³³⁷ periods (see Fig. 8). The differences between modelled and official maps are partly due to
³³⁸ the topography of the Po floodplain, which is not reproduced in the STRT used by both
³³⁹ JRC and CA2D maps. Indeed, the area enclosed by the main levees has a complex system
³⁴⁰ of minor embankments, which are designed for lower flood return periods than the main
³⁴¹ levees (Castellarin et al., 2011). This explains why AdBPo maps are quite similar for return
³⁴² periods of 20-50 years and 100-200 years (see Figure 8).
³⁴³ The narrow extent of flooded areas for return periods of 50 and 100 years in sectors of the
³⁴⁴ river network suggests that the channel conveyance may be overestimated in CA2D maps.
³⁴⁵ However also our reference AdBPo maps show very similar flood extents for return periods
³⁴⁶ of 20-50 and 100-200 years as explained above, therefore the CA2D underestimation can not
³⁴⁷ be quantified. Future work will anyway refine the methodology of channel "digging". This
³⁴⁸ is indeed an open research question, due to the absence of large-scale methods or datasets
³⁴⁹ to estimate river channel depth (Dottori et al., 2016d). Nevertheless, it is worth noting that
³⁵⁰ the method presented here improves the sensitivity to return period of flood extent maps.
³⁵¹ Conversely, JRC maps calculated for different return period have limited differences, due
³⁵² to the absence of river geometry details. These results confirm that the inclusion of a river
³⁵³ channel network is necessary to guarantee acceptable results in the simulation of flood depths
³⁵⁴ and extent for all return periods (Neal et al., 2012).

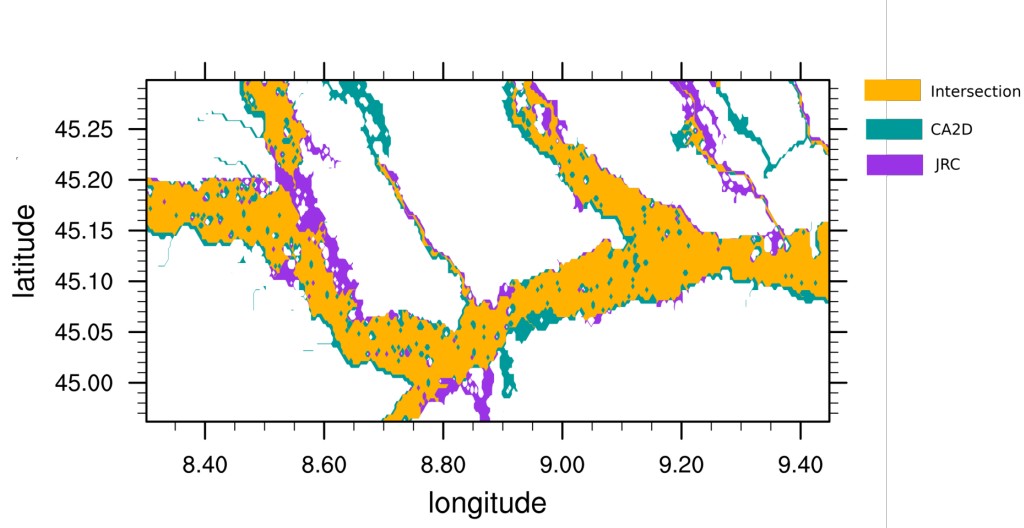

Figure 9: CA2D and JRC flood hazard maps for the 500 years return period

# 4  Conclusions

In this paper we investigate the feasibility of producing high-resolution flood maps using an innovative approach which reshapes the digital elevation models by simulating a "digging" assuming that no floods take place for discharges associated to the return period of 1.5 years, representing the conveyance capacity of the river channel. The main purpose of this method development is to be able to apply it also in those regions where there are no available information about river natural and man-made banks. A 2-dimensional hydraulic model is used to simulate the propagation of the hydrographs across the HydroSHEDS void filled DEM, which was processed to yield an estimate of bankfull discharge. The evaluation of the produced flood maps was performed through some case studies of observed flood extent satellite data, and through existing flood maps over the entire domain, showing a good spatial agreement with observations for high return periods. Comparison for lower return periods showed that the DEM-reshaping method improves the sensitivity to return period of flood extent maps but needs further improvement, for instance, combining observed data about river bed depth and width and discharge (Yamazaki et al., 2014). The validation of the method in a region where all the hydrological and hydraulic information are available will allow us to extend the method elsewhere.



## 5   Acknowledgments

The authors gratefully acknowledge the financial support of the Allianz Insurance Company
for the realization of this project.

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
