# Peer review of "An integrated hydrological and hydraulic modelling"

_Natural Hazards and Earth System Sciences, 2019_

## Referee Comment (RC1) · Dominik Paprotny (Referee) · 6 Jan 2020

The manuscript "An integrated hydrological and hydraulic modelling approach for the flood risk assessment over Po river basin" presents a chain of models for producing flood hazard maps in northern Italy. The paper is interesting and touches an important topic, but has many issues in terms of writing, analysis and underlying modelling work. Firstly, I will list my major concerns in the order of appearance in the paper. Then, I will list some minor comments and suggestions to improve the text.

Introduction: the introduction gives little information on the research gaps that exist and which the authors are trying to address. No research goals are stated, and no clue is

given what innovation or contribution to the field is introduced by the paper. Most of the introduction is a general overview of history of flood modelling, though interesting by itself, is mostly not too relevant for the study, and it barely cites any literature from the past 10 years. A revised introduction should clearly state what the study contributes and which research gaps it addresses, and the literature overview should be focused on those aspects, and cite more recent papers given the enormous developments in the field in the past decade. Also, it should be explained what is the existing flood hazard map availability for Italy and why new maps are needed (especially since, as described later in the paper, national maps are already available!).

Methods (2.1): the main innovations here, i.e. the use of a new precipitation dataset and a new implementation with a high-resolution DEM, are very briefly described. The resolution of the DEM and precipitation data should be clearly written, as should the information about other necessary inputs for hydrological modelling (evapotranspiration, snowmelt, infiltration etc.), model set-up (e.g. timestep) and model calibration.

Methods (2.3): from the text it seems that the only change in the hydraulic model is the parallelization. This should be clearly written, and more details should be provided as this is an important addition. It would be particularly useful to describe to what simulation set-up Fig. 3 pertains to. If other changes were made to the model they need to be described.

Methods (2.4): most of all, the method "digging" the channel in the DEM is not well described. Was the bankfull depth used directly for the lowering grid cells in the 3" resolution DEM, or the resolution of the DEM (which is coarser than width of most rivers in the study area) was accounted for by reducing the depth accordingly to achieve the same wetted perimeter? If the former, then the conveyance of the rivers will be vastly overestimated and needs to be corrected. If the latter, then it needs to be properly described. Also, what does the "ad hoc" re-shaping of HydroSHEDS in the abstract actually refer to? Further, there is too little information about the simulation set-up, such as timestep, spin-up time, simulation time, calibration procedure (if there was

any) or roughness coefficient selection (was it spatially-variable? was it adjusted by calibration?).

Validation (3.1): the validation mentions "tuning" the model (line 264), but no information about calibration procedure were provided. Also, the text mentions "Due to the relatively small size of the simulated domains, the duration of all flood simulations was set to 240 h" (line 266). Does this refer to simulation domains from Figure 1? Or the sub-domains mentioned in the previous section? (line 259). This should be clarified to avoid confusion. Still, if the simulation was done over the whole Po river, isn't 240 hours far too low to capture the response of the catchment (I made a quick check with an empirical equation, which suggests so)?

Validation (3.2): the validation here is only visual, but as Figure 7 shows flooded areas extracted already from the satellite images, it would be possible to apply the method of comparing flood maps from section 3.3 to compute the different indices. Also, the impression of good match between the modelled and observed flood extents partially stems from showing a 500-year flood map for comparison, instead of only 100-year flood. Finally, given that the hydrological simulation made by the authors cover the time of the event, wouldn't it be a better comparison by running the hydraulic model specifically for the 2016 event?

Validation (3.3): official hazard maps are used here for comparison, but no information how they were produced are provided. This is important in order to assess the source of differences with the authors maps. Also, the authors only show the results for a 500-year flood map, while discussing other return periods as well. Those results should be shown. Especially as authors claim there maps being better than JRC's , but the results for the 500-year maps are actually worse. Also, it is well possible that the authors' models underestimate flood hazard severely – but that could be made clearer by information how the reference Italian maps were made. Also, the JRC maps might not include channel geometry, but account for this by removing mean discharge from the design hydrographs, as I did in my pan-European flood modelling work, too (Paprotny

et al. 2017, cited by authors). Other researchers (Ward et al., 2013; Sampson et al., 2015) accounted for this by removing 2-year discharge.

Finally, there is no discussion section in the paper, hence missing many important aspects. Uncertainties and limitations are not discussed (e.g. related to the channel "digging" or design hydrographs). Ways to further improve the work and next steps are not discussed too, and neither is relevance of the work for making projections of flood hazard under climate change. But most importantly, the issue of flood protection is ignored. Though the authors write that the channel "digging" accounts for "man-made" banks, but a return period of 1.5 years is below even the most meagre flood defences. In practice of flood hazard modelling, assumptions about the level of flood protection has very strong influence on the results, as I show in Paprotny et al. (2017). Without this, any improvements to the hydrological or hydraulic modelling are mostly lost. If this is not addressed by the authors in their model, it needs to be at least extensively discussed.

Minor comments:

Title: the authors write "integrated hydrological and hydraulic modelling" but in reality the two are run entirely separately. Also, the work relates to flood hazard mapping in the Po river basin and not "flood risk assessment" (risk is not addressed by the paper) "over" Po river basin. The authors should propose a new title that includes only items that are covered by the paper.

L14: typo "90m"

L26: should be "are" not "and"

L37: "Flood Risk Management Directive" is not an official name, hence it should be refer to as "Floods Directive" in parentheses.

L52: "For limited area gauged basins" is not understandable, probably should be "For small, gauged basins"

L110-111: a large-scale map, in geography, covers a small area (large amplification). Mixing "scale" as in maps, and "scale" as in process is commonplace and should be corrected to "...assumes that flood hazard maps over a large domain can be derived from an ensemble of smaller sub-simulations..."

L122: "D8" should be explained.

Figure 1: the map lacks legend, grid or scale. Also, the source of the underlying map should be identified in the caption.

Section 2.2: throughout, authors use multiple letters for a single variable. A single letter should be used e.g. $S$ instead of $SDH$. Subscripts could be also used instead to differentiate.

L172 and subseq.: it should be clearly marged that eq. 3-5 are directly taken from Maione et al. (2003).

179-L180: the authors mention and show equation for the falling limb, but shouldn't there be also an equation for the rising limb of the hydrograph?

L186: what method was used for fitting?

L191: write specifically which station.

Figure 3 and others: the size of figures, their labels and general appearance should be synchronized throughout the paper, as at the moment that give a very messy appearance together.

L232: "for larger domains" not "on large scales".

L247-L249: this is the part where it is particularly unclear how the "digging" was made.

L274: the results referred to by authors were actually presented in [Paprotny D., Morales Nápoles O. (2017) Estimating extreme river discharges in Europe through a Bayesian Network. Hydrology and Earth System Sciences 21, 2615–2636.] rather

[Figure]

than in Paprotny et al. (2017) cited here.

Figure 6: a 1:1 line should be added to the graph.

Figure 7: "meters" is missing in the lower legend.

Section 3.3: again, a single letter (with possible subscript) should be used per variable.

L321: Alfieri et al. (2014, 2015) should be cited here regarding the methodology of the JRC maps.

L338: typo "STRT".

L351-352: rather due to lack of flood defences in the model

L361: as noted above, the authors do not really account for "man-made banks", if those are formed by flood defences.

L363-364: authors write that "[t]he evaluation of the produced flood maps was performed through some case studies of observed flood extent", but actually only one case study is shown (Nov. 2016).

---

## Referee Comment (RC2) · Alex Curran (Referee) · 19 Jan 2020

**Review: An integrated hydrological and hydraulic modelling approach for the flood risk assessment over Po river basin.**

**General Comments**

This paper presents very interesting work on the topic of flood risk assessments using novel hydraulic and hydrological modelling techniques. The paper is well written, referenced and structured. The results are presented clearly, and are honest (perhaps even modest) about their performance. However, while the methods are innovative, the major contribution developed is described as "an innovative approach which reshapes the digital elevation models". Despite this being the crux of the work, this digging method is not described at all, other than calling it an 'ad-hoc' process in the abstract.

The resolution used was 90m, and the main purpose of the method is to "apply it also in those regions where there is [limited information]". In regions with limited information, river channels are very likely to be less than 90m wide, so this raises some difficult questions about the 'digging' process. Due to the lack of explanation on this vital component, I am suggesting this paper needs 'major revisions' before it can be accepted. However, in reality, it is only this point that needs to be revised, rethought and clearly explained. For example, a simple fix would be to conclude the innovation of the work as a 'an innovative combined hydraulic and hydrological modelling process' which shows good results. In this case, however, the computational efficiency and performance advantages over other methods (such as JRC) should be clearly explained.

Some smaller specific comments are given below.

**Specific Comments**

Title: "…over THE Po River basin".

Page 1, Line 27: "the results ARE less satisfactory…"

P2 L37: "which mandate is" (English)

P2 L41: "for A few regions of the globe"

P2 L70: I am not sure if these drawbacks are relevant. 1D schemes are rarely described as computationally expensive. Also, the fact that areas between cross-sections are not represented is true of any model that uses discretisation (i.e. all numerical models). A 2D model also discretises an area into uniform blocks that don't represent variation within the blocks.

P4 L122: I assume D8 relates to the deterministic eight nodes method (Martz and Garbrecht 1992), but this should be stated and referenced.

P4 L125: No mention is made as to how CHyM handles evapotranspiration.

P4 L130: Although 9 domains were simulated in the overall project work done, I don't think it is relevant to the story here, which focuses on the Po / Tanaro. This is also true for Figure 1.

P4 L131: "The domains are matching the…" I think 'match' is better here, but in general the sentence is a bit long and awkward, and could be rethought.

P6: L170: "(flatter) rising (falling) limbs of..." The authors are trying to be less verbose here, but the resulting is just confusing. Perhaps 'vice-versa'?

P6 L176: Even though I am familiar with Maione's work, this description was hard to follows. Perhaps it could be explained with reference to the diagram below;

[Figure]

P7 L196: Apologies if I am incorrect here, but as I understand it, CA hydraulic modelling is simply finite volume modelling in which the volumes are balanced over a group of neighbouring cells for each cell, rather than over the whole domain. Given that the definition of a cellular automaton is (basically) a set of rules for a domain of grid cells, surely all 2d hydraulic models could not be called 'cellular automata', and the name is not required.

P7 L208: "..., as expected, ..." or "...(as expected)..."

P9 L 213: interested river network? This sounds strange

P9 L214 "the following section" is ambiguous here, as it does not refer to the actual next (sub)section 2.4.

P9 L242: As mentioned above, does a 90m DEM allow for this digging? What is the 'ad-hoc' process? Does this work-around mean the process is only suited to the presented application?

P10 L268: The SDHs are built using observed data, so how exactly does using a model validate them?

P10 L272: This should be 'area under the curves' However, given this is Q for a set period, I think it would be better to simply use 'total volume' as the metric.

P12 Figure 7: This figure is poor. The elevation legend has no units, and has too large a range. The flood legends are similar in both the aerial and GIS images, but represent different things.

P12 L287: Where is it catalogued as such? 100 year rainfall or 100 year level at a certain location?

P13 L294: I think the problems of SAR should be mentioned (double-bounce etc.) as they can be seen in the satellite image.

P13 L317: How are the AdbPo maps produced?

P15 L328: This part raises more questions about the dug channels. Why was 1.5 years chosen? Why not do a second 'dig' in the areas between main levees (or minor levees) to the same conveyance as the protection level. In other words, why not dig a conveyance of 200yrs in the area of Fascia B?

P16 Conclusion: As Maione's RP method is used in the overall combined method presented, it is limited to producing flood maps, which should be mentioned. Furthermore, such maps don't allow for spatial homogeneity issues (for example, a 100yr event doesn't produce a 100yr level at all locations).

P16 L360: The fact that the method is developed for regions with limited information seems to be mentioned here for the first time.

P16 L362L: Which hydrographs?

---

## Referee Comment (RC3) · Anonymous Referee #3 · 6 Feb 2020

An integrated hydrological and hydraulic modelling approach for the flood risk assessment over Po river basin, by Nogherotto et al. – submitted to NHESS Overall evaluation: The aim of the manuscript is to provide a combined approach for mapping flood hazard for different return periods, starting from a precipitation dataset and using a statistical procedure to design synthetic hydrographs, to be then used to simulate inundation scenarios in the floodplain.

Although the topic is interesting, the manuscript is well written and structures and the its scope is in line with the journal, I don't find this study innovative enough to be published in NHESS. The methodology, as described in the manuscript, is the one developed in

Maione et al. (2003) and used to map European flood hazard in Dottori et al. (2006), who also use the identical approach for simulating the flooding dynamics, with the only difference in reducing the flood hydrograph discharges by subtracting the estimated average daily discharge, instead of "digging" the DTM as stated in this manuscript. The main differences declared in this study are the different database and DTM used that just characterize an application of the cited methodologies on different case studies. This is the main reason why I would reject the manuscript in its current form. Some other concerns about the manuscript are listed below. I hope the authors will find them useful and I encourage them to resubmit a thoroughly and carefully revised version of the present study, clearly specifying the innovations made.

Major comments:

Title: I don't think this can be defined an "integrated" approach, because the two models (hydrological and hydraulic), as I understand from the description, run separately. I would relate to a "combined" approach instead. In addition, flood risk in literature is defined as the combination of flood hazard, exposure and vulnerability, while the manuscript refers to hazard only.

Introduction: I would add a part about flood hazard maps, investigating how they are currently designed, which is the situation in Italy (where you focus your study), etc.

Section 2.3: I find this part too short and less detailed, it becomes clearer only after reading the paper of Dottori et al. (2016). Details about boundary conditions, roughness coefficients and other useful details for modelers should be added. In the last part of the section, the reader is expected to find how the following sections are organized, but the list of the steps here stated don't agree with the chapters.

Section 2.4: the main lack here, and in the whole manuscript, is the description of the digging method. Being one of the few modifications of the cited methodologies, it needs to be clearly described (included the motivation of this choice instead of following Dottori's methodology), in order to justify the application of this approach. Again, details

need to be added, such as how the model works in the external areas, how the different flood maps of the virtual stations are merged, levees breaching mechanisms (if any), etc.

Case studies and study areas: They need to be better described and motivated, included the choice to refer to three different studies areas. "Upper Po basin", "the area in the south of Turin", "the area of Alessandria" are too general, please specify where they are (also with Figures), how big they are, etc. Is the area in Figures 8 and 9 the same as in Figures 4 and 5? If yes, why is it cut in the northern and southern part? If not, why not?

Section 3.1: Please pay more attention in the terminology used: the SDHs cannot be validated using observations from the gauging stations for Tr 50, 100 and 500 years. . . there are not observations for 500 years return period!!! These values are extrapolated from statistical studies starting from observation, but it needs to be clarified. In addition, I would not say "tuning" the model, because the model was already developed by Maione et al. (2003), maybe it was applied to the new data. If there are substantial modifications instead, please clarify it, because it is not evident up to now.

Section 3.2: Why do authors refer also to Tr = 500, when they write that the November 2016 event was catalogued as a 100years return period event? In addition, I think that the sentence "We can see that the observed event, associated to a return period of 100 years, is fairly good represented by the model (Fig. 7 (b) and (d)) as the maps include the particular events observed" cannot be accepted in a scientific manuscript as a valid result of a study. The judgment of the validity of the approach cannot be based on the impression of the reader that looks at the two maps and conclude that they are similar! Why authors didn't use the same indices as in Section 3.3? I would provide a initial Section in Chapter 3 where describing the indices used, and then perform all the comparisons using this indices.

Section 3.3: In order to better understand and discuss the results of the study, an explanation on why authors chose River Po Basin Authority and JRC maps as comparison must be added, and also how they are derived. Why authors say that it is possible to calculate indices only for Tr = 500 years but then show and comment results also for Tr = 50 and 100? It is not clear, in addition and related to this comment, which models consider the embankment system. Po river has an important levee system, which has a very important influence on the results of hydraulic simulations (see, e.g., Wing et al. (2019)*). This issue must be considered at least in the discussion result. In addition, there are no reference at water depth results in the study, that's why the sentence at P. 15 L. 352-354 is not correct.

In general, results in Section 3 have to be deeper investigated (e.g. in Figure 9, the explanation of the differences between CA2D and JRC maps are not taken into consideration at all).

Figures 8 and 9: I find the way to visualize results in Fig. 8 very unrepresentative. I would, instead, represent maps in Figure 8 as in Figure 9, in terms of comparison of CA2D and JRC maps, respectively, with AdB maps, following the representation of results in Table 1. The same for Figure. 4.

Minor comments:

P. 1 L. 26: results ARE less satisfactory. . .

P. 2 L. 36: the development of flood hazard maps is only one of the mandates of the European Flood Directive (better than "European Union Flood Risk Management Directive")

Among 2d models, I would mention also the 2d version of Hec-Ras, very used in the last years.

Figure 1: this is unnecessary, it doesn't add anything interesting to the study.

P. 6 L. 163: the right reference is Alfieri et al. (2014) instead of 2013, if I understand what authors refer to.

P. 7 L. 211 – P. 9 L. 213: Please rephrase, it is not clear.

P. 9 L. 213-214: the following Section is 2.4, but this is not the section in which the results are shown. Please correct.

Figure 4: y-axis label is missing.

Figures 4, 5, 7, 8, 9: scale bar and north arrow are missing.

P. 10 L. 266: add reference

P. 10 L. 267: "small size": please specify quantitatively.

P. 10 L. 267-268: please explain better, with references.

Please provide reference for "observational data" in the Po River.

Figure 5: Please specify in the caption which model is used to map hazard areas.

Figure 6: a) and b) labels are not shown in the figure.

Figure 7: what does the legends refer to?

P. 13 L. 300: perform a COMPARISON BETWEEN existing. . .

P. 13 L. 314: I would use the terminology "CA2D maps" from the beginning of the manuscript.

P. 13 L. 315: PO RIVER BASIN AUTHORITY instead of River Po Authority.

P. 15 L. 338: the SRTM used. . .

P. 16 L. 362: which hydrographs?

P. 16 L. 365: the comparison is not in the ENTIRE domain, please clarify

* Wing, O., et al., A New Automated Method for Improved Flood Defense Representation in Large‐Scale Hydraulic Models, Water Resources Research, 55 (12), 11007-11034, 2019. https://doi.org/10.1029/2019WR025957

---

## Author Comment (AC1) · 16 Apr 2020

Please see the attached file with the comments from the Referee and the author's response, and the file with author's changes in manuscript.

Please also note the supplement to this comment: https://www.nat-hazards-earth-syst-sci-discuss.net/nhess-2019-356/nhess-2019-356-AC1-supplement.zip